# Improvement in Thermal Stability of Flexible Poly(L-lactide)-*b*-poly(ethylene glycol)-*b*-poly(L-lactide) Bioplastic by Blending with Native Cassava Starch

**DOI:** 10.3390/polym14153186

**Published:** 2022-08-04

**Authors:** Yaowalak Srisuwan, Yodthong Baimark

**Affiliations:** Biodegradable Polymers Research Unit, Department of Chemistry and Centre of Excellence for Innovation in Chemistry, Faculty of Science, Mahasarakham University, Mahasarakham 44150, Thailand

**Keywords:** poly(lactic acid), block copolymer, cassava starch, biocomposites, thermal stability

## Abstract

High-molecular-weight poly(L-lactide)-*b*-poly(ethylene glycol)-*b*-poly(L-lactide) triblock copolymer (PLLA-PEG-PLLA) is a promising candidate for use as a biodegradable bioplastic because of its high flexibility. However, the applications of PLLA-PEG-PLLA have been limited due to its high cost and poor thermal stability compared to PLLA. In this work, native cassava starch was blended to reduce the production cost and to improve the thermal stability of PLLA-PEG-PLLA. The starch interacted with PEG middle blocks to increase the thermal stability of the PLLA-PEG-PLLA matrix and to enhance phase adhesion between the PLLA-PEG-PLLA matrix and dispersed starch particles. Tensile stress and strain at break of PLLA-PEG-PLLA films decreased and the hydrophilicity increased as the starch content increased. However, all the PLLA-PEG-PLLA/starch films remained more flexible than the pure PLLA film, representing a promising candidate in biomedical, packaging and agricultural applications.

## 1. Introduction

Among bioplastics, poly(L-lactic acid) or poly(L-lactide) (PLLA) is a promising substitute for traditional petroleum-based plastics in many fields, such as biomedical, tissue engineering, drug delivery and packaging applications [1,2,3,4]. This is due to its excellent biodegradability, biocompatibility and compostability, as well as being environmentally friendly and having good processability and feasibility for an increased production scale [5,6]. However, the low flexibility of PLLA, due to its high glass-transition temperature (T_g_, around 60 °C), limits its use in many applications [7,8,9]. High-molecular-weight PLLA-*b*-poly(ethylene glycol)-*b*-PLLA triblock copolymers (PLLA-PEG-PLLA) have exhibited more flexibility than PLLA because of the high flexibility of PEG middle blocks [10,11,12].

Unfortunately, these pure PLLA-PEG-PLLAs had high melt flow ability (low melt strength), which is not appropriate for many processing applications, such as injection molding, blow film molding and extrusion molding [11,12]. The melt flow properties of PLLA-PEG-PLLA can be controlled by reaction with a chain extender to form long-chain branching structures. However, our previous works reported that the thermal stability of PLLA-PEG-PLLA decreased after the chain extension reaction [11,12,13]. The branching structures of the chain-extended PLLA and PLLA-PEG-PLLA suppressed its thermal stability by preventing molecular interactions. This resulted in a narrower window of processing for the chain-extended PLLA-PEG-PLLA (the range between its melting and thermal decomposition temperatures was narrower). The 60/40 chain-extended PLLA-PEG-PLLA/thermoplastic starch (TPS) blends were co-continuous phase structures, indicating that they had good phase compatibility [14]. The PEG middle blocks enhanced the phase compatibility between PLLA end blocks and TPS phases. Moreover, the TPS blending improved the thermal stability of PLLA-PEG-PLLA. However, the effect of native starch on the thermal stability of PLLA-PEG-PLLA has not yet been explored in detail.

Native starches are low-cost and non-scarce biopolymers and have been blended with various biodegradable plastics, such as PLLA [15,16,17], poly(ε-caprolactone) [18] and polyhydroxybutyrate [19], for reducing their production costs and for maintaining their biodegradability. However, the highly brittle PLLA/native starch composites are limited in practical applications due to the poor mechanical properties of native starch and poor phase compatibility between PLLA and native starch particles [15,20]. Therefore, the aim of this work is to investigate the influence of native cassava starch (5, 10 and 20 wt%) on the thermal stability, phase morphology, mechanical properties and hydrophilicity of PLLA-PEG-PLLA compared to PLLA/native cassava starch blends.

## 2. Experimental Section

### 2.1. Materials

PLLA and PLLA-PEG-PLLA were synthesized through ring-opening polymerization of L-lactide monomer (96% L-enantiomer content) at 165 °C in bulk under a nitrogen atmosphere, as described in our previous works [11,12,14]. Stannous octoate (95%, Sigma, St. Louis, MO, USA) was used as a catalyst. Moreover, 1-dodecanol (98%, Fluka, Buchs, Switzerland) and poly(ethylene glycol) with molecular weight of 20,000 g/mol (Sigma, St. Louis, MO, USA) were used as initiators for the synthesis of PLLA and PLLA-PEG-PLLA, respectively. Number-average molecular weight (M_n_) and dispersity index (*Đ*) of PLLA, analyzed by gel permeation chromatography (GPC, e2695 separations module, Waters, Milford, MA, USA), were 88,400 g/mol and 2.3, respectively, while the M_n_ and *Đ* values of PLLA-PEG-PLLA were 89,900 g/mol and 2.8, respectively. Melt flow index (MFI) of obtained PLLA was 24 g/10 min at 190 °C under 2.16 kg load. However, the MFI of obtained PLLA-PEG-PLLA was too high. The melt flow property of PLLA-PEG-PLLA was then adjusted to be close to the value for PLLA obtained by chain extension reaction with 4 parts per hundred of resin by weight (phr) of Joncryl^®^ ADR4368 chain extender (BASF, Bangkok, Thailand) [11,14]. The MFI of the obtained chain-extended PLLA-PEG-PLLA was 23 g/10 min. Native cassava starch (food-grade, 22% amylose content) was supplied by Kriangkrai Co., Ltd. (Nakornprathom, Thailand). A scanning electron microscope (SEM, JSM-6460LV, JEOL, Tokyo, Japan) image of native cassava starch is presented in Figure 1. The starch particles were in the range of 5–20 µm.

### 2.2. Preparation of PLLA/Starch and PLLA-PEG-PLLA/Starch Composites

Chain-extended PLLA-PEG-PLLA and cassava starch were dried in a vacuum oven at 50 °C for 24 h before fabricating the composites. PLLA-PEG-PLLA/starch composites were prepared using a Rheomix batch mixer (HAAKE Polylab OS, Thermo Fisher Scientific, Waltham, MA, USA) at 180 °C with a rotor speed of 100 rpm for 5 min. The composites with PLLA-PEG-PLLA/starch ratios of 100/0, 95/5, 90/10 and 80/20 (*w*/*w*) were investigated. PLLA/starch composites were also prepared using the same process for comparison.

Before film forming, the composites were dried at 50 °C in a vacuum oven for 24 h. Composite films (0.2 mm in thickness) were prepared using a compression molding machine (Auto CH, Carver, Inc., Savannah, GA, USA). The composite pellets were filled into a mold and preheated at 180 °C for 3 min without compression force, followed by hot pressing at the same temperature under 5.0 MPa load for 1 min. Afterwards, the mold was rapidly cooled in a water-cooled press under 5.0 MPa load for 1 min. The obtained film samples were stored in a desiccator for 24 h before characterization.

### 2.3. Characterization of PLLA/Starch and PLLA-PEG-PLLA/Starch Composites

Thermal transition properties of the samples were investigated using a differential scanning calorimeter (DSC, Pyris Diamond, PerkinElmer, Waltham, MA, USA). First, samples were heated at 200 °C for 3 min to erase the thermal history, before fast quenching to 0 °C and heating from 0 °C to 200 °C at a rate of 10 °C/min under a nitrogen gas flow. The degree of crystallinity (*X*_c_) was calculated according to the following equation:*X*_c_ (%) = [(ΔH_m_ − ΔH_cc_)/(93.6 × *W*_PL__LA_)] × 100(1)
where ∆H_m_ and ∆H_cc_ are enthalpies of melting and cold crystallization, respectively. Moreover, 93.6 J/g is the theoretical ΔH_m_ for 100% crystalline PLLA [14]. *W*_PL__LA_ is the PLLA weight fraction of the samples calculated from PLLA fractions (PLLA = 1.00 and PLLA-PEG-PLLA = 0.83 obtained from ^1^ H-NMR [11]) and the starch content.

Thermal decomposition behaviors of the samples were determined using a thermogravimetric analyzer (TGA, SDT Q600, TA Instruments, New Castle, DE, USA). Samples (5–10 mg) were heated from 50 °C to 800 °C at a rate of 20 °C/min under a nitrogen flow.

An SEM (JSM-6460LV, JEOL, Tokyo, Japan) operated at 20 kV was introduced to observe the phase morphology of the film samples. The films were cryogenically fractured after immersion in liquid nitrogen for 10 min and were sputter-coated with gold before SEM analysis. Film morphology of the starch-free composite films was also investigated after the cryogenically fractured films were immersed in 6 N HCl aqueous solution at room temperature for 3 h to remove the starch phase [21].

The tensile testing of film samples (100 mm × 10 mm) was carried out using a universal mechanical testing machine (LY-1066B, Dongguan Liyi Environmental Technology Co., Ltd., Dongguan, China) with a 100 kg load cell according to ASTM D882. A crosshead speed of 50 mm/min and a gauge length of 50 mm were used. The tensile properties were averaged from at least five measurements.

Water contact angle of each film surface was recorded using a contact angle goniometer (Ramé-Hart Instrument Co., Succasunna, NJ, USA) after 15 s. Each sample was averaged over five different determinations.

Moisture uptake of film samples (20 mm × 20 mm) was investigated as follows. The film samples were weighed after drying at room temperature in a vacuum oven for 48 h. They were kept in a desiccator with 90 ± 5% relative humidity, maintained with a saturated sodium chloride solution at 25 °C. The sample films were weighed again after being kept in the desiccator for an interval of time. The moisture uptake of film samples was calculated from Equation (2). The averaged moisture uptake was obtained from five different determinations [14].
Moisture uptake (%) = [(M_f_ − M_i_)/M_i_] × 100(2)
where M_i_ and M_f_ are the weights of the film before and after the test, respectively.

## 3. Results and Discussion

### 3.1. Thermal Transition Properties

The thermal transition properties of the composites were determined by DSC, and the DSC heating curves of PLLA/starch and PLLA-PEG-PLLA/starch composites are given in Figure 2. The DSC results are summarized in Table 1. The T_g_, T_cc_ and T_m_ values of PLLA and PLLA-PEG-PLLA matrices did not change significantly when the starch was blended and the starch content was increased, indicating the immiscibility of these matrices and native starch [18]. The *X*_c_ values of the pure PLLA (*X*_c_ = 23.6%) and pure PLLA-PEG-PLLA (*X*_c_ = 15.3%) slightly increased to 27.1% and 20.3%, respectively, as the 5 wt% starch was blended. Further increase in starch content in the composites led to a decrease in *X*_c_. This indicates that the presence of starch particles as a minority fraction enhanced the nucleation of PLLA-based matrices at the particle interface [18,22,23].

However, starch particles with higher starch content (10 and 20 wt%) may aggregate, leading to reduced nucleation efficiency [18]. It should be noted that the *X*_c_ values of the PLLA-PEG-PLLA-based composites with 10 wt% (*X*_c_ = 19.4%) and 20 wt% (*X*_c_ = 16.1%) starch content were still larger than those of the pure PLLA-PEG-PLLA (*X*_c_ = 15.3%), whereas the *X*_c_ values of both the PLLA/starch composites with 10 wt% (*X*_c_ = 22.6%) and 20 wt% (*X*_c_ = 18.7%) starch content were lower than those of the pure PLLA (*X*_c_ = 23.6%). The results suggested that the phase compatibility between PLLA-PEG-PLLA matrices and starch particles was better in terms of the nucleation effect [24,25]. The dispersed component could promote the crystallization of the crystalline matrix due to the nucleation activity of the matrix/starch interface [26]. The dispersed starch particles were well compatible with the PLLA-PEG-PLLA matrices on matrix/starch interfaces and induced a nucleation barrier lower than that in pure PLLA-PEG-PLLA to accelerate the crystallization rate of PLLA-PEG-PLLA. In addition, blending of native starch particles in this work revealed a better nucleation effect for the PLLA-PEG-PLLA matrix compared to the blending of TPS in our previous work [14]. The *X*_c_ values of PLLA-PEG-PLLA/TPS blends steadily decreased as the TPS content increased.

### 3.2. Thermal Decomposition Behaviors

The thermal decomposition behaviors of the samples were determined from their TG thermograms, as shown in Figure 3. The pure PLLA showed a single step of thermal decomposition in the range of 250–450 °C. The TG thermogram of native cassava starch had two steps of thermal decomposition in the ranges 50–150 °C and 250–500 °C, as presented in Figure 4, due to the evaporation of residue moisture and pyrolysis of starch, respectively [24]. The decomposition temperature at 5% weight loss (5%-T_d_) of pure PLLA was at 291 °C, as reported in Table 2. The 5%-Td values of PLLA/starch composites shifted to lower temperatures as the starch content increased due to the evaporation of moisture from the starch. The native cassava starch in this work had around 10 wt% residue ash at 800 °C. The TG thermograms of PLLA/starch composites in Figure 3 (above) also exhibit a single step of thermal decomposition in the range 250–450 °C.

The pure PLLA-PEG-PLLA showed two steps of thermal decomposition in the ranges 250–350 °C and 350–450 °C due to thermal decomposition of PLLA and PEG blocks, respectively [11,12]. The 5%-T_d_ of PLLA-PEG-PLLA shifted from 286 °C to 278 °C when the 5% starch was blended. However, the 5%-T_d_ of PLLA-PEG-PLLA/starch composites shifted to higher temperatures when the starch content increased up to 10 and 20 wt%. It could be clearly seen that the thermal decomposition steps of PLLA end blocks dramatically shifted to higher temperatures as the starch content increased, as shown in Figure 3 (below). This indicates that the blending of native starch effectively improved the thermal stability of the PLLA end blocks of PLLA-PEG-PLLA, similar to the blending of TPS [14]. The residue ash in both the PLLA/starch and PLLA-PEG-PLLA/starch composites increased steadily with the starch content, as summarized in Table 2.

The derivative TG (DTG) thermograms in Figure 5 also enabled the determination of the thermal decomposition behaviors of composite samples. The DTG peak was assigned to the temperature of maximum decomposition rate (T_d,max_) of the composites, as is also reported in Table 2. The pure PLLA and all the PLLA/starch composites had a single T_d,max_ peak in the range 365–369 °C, attributed to the “unzipping” mechanism at the chain ends of PLLA [27]. The native cassava starch had a T_d,max_ peak at 327 °C, as assigned in Figure 4. The small T_d,max_ peaks of starch minor fractions may be overlapped with the large T_d,max_ peaks of PLLA major fractions. Thus, the addition of starch did not significantly change the thermal decomposition behaviors of the PLLA matrix.

The pure PLLA-PEG-PLLA had two T_d,max_ peaks at 319 °C and 422 °C, attributed to the thermal decomposition of PLLA (PLLA-T_d,max_) and PEG (PEG-T_d,max_) blocks, respectively [11,27]. It was found that the PLLA-T_d,max_ peaks of all the PLLA-PEG-PLLA/starch composites (336–363 °C) were at higher temperatures than that of the pure PLLA-PEG-PLLA (PLLA-T_d,max_ = 319 °C). The PLLA-T_d,max_ peaks of the composites dramatically shifted to higher temperatures as the starch content increased. However, the PEG-T_d,max_ peaks of the composites were in the range 418–422 °C and did not shift significantly with starch blending. This may be due to the starch fraction being completely decomposed before the thermal decomposition of PEG.

The TG and DTG results indicated that the interactions between PLLA-PEG-PLLA and starch were stronger than those between PLLA and starch and suggested that the starch acted as a low-cost and effective thermal stabilizer for PLLA-PEG-PLLA. This may be explained by the formation of hydrogen bonds between the oxygen of PEG middle blocks and hydroxyl groups of starch [14,25,28]. In addition, the thermal stability of compatible polymer/starch blends was improved by products from the thermal decomposition of starch [29,30].

### 3.3. Phase Morphology

SEM images of the composite films were used to observe their phase morphology, as shown in Figure 6. Both the pure PLLA and PLLA-PEG-PLLA films illustrated in Figure 6a,e, respectively, had no phase separation. For all the PLLA/starch and PLLA-PEG-PLLA/starch films in Figure 6b–d and f–h, respectively, starch particles were found to be dispersed in the film matrices in a typically heterogeneous structure. Micro-voids on the film matrices occurred from the detachment of starch particles during the cryo-fracture step, which suggested poor interfacial adhesion between the film matrix and starch particles [25,31]. It appears that the number of micro-voids (or starch particles) increased with the starch content. It could be seen that there were more adhered starch particles for PLLA-PEG-PLLA/starch films than on the PLLA/starch films for the same starch content; this supported the hypothesis of stronger interactions between PLLA-PEG-PLLA and starch, corresponding to the TG/DTG results, as described above [28].

Figure 7 presents SEM images of the cryogenically fractured 80/20 composite films after selective etching with HCl solution to remove the starch phases. Many micro-voids from the removal of the starch particles were obtained on the film matrices. This confirms that phase separation between film matrices and dispersed starch particles occurred for both the PLLA/starch and PLLA-PEG-PLLA/starch films. It should be noted that the film matrix of the 80/20 PLLA/starch composite was cracked, as indicated by the white arrows in Figure 7 (above), but the 80/20 PLLA-PEG-PLLA/starch film in Figure 7 (below) was not cracked. This implied that the 80/20 PLLA/starch film was brittle because the starch content was too high and also supported the supposition that interactions between PLLA and starch were poor.

### 3.4. Tensile Properties

Figure 8 shows tensile curves of the composite films, and the tensile results were averaged and are summarized in Table 3. Addition of starch decreased all determinations of tensile stress, strain at break and Young’s modulus for the PLLA films. This was due to the poor mechanical properties of starch and poor phase compatibility between hydrophobic PLLA and hydrophilic starch [15,20]. The 80/20 PLLA/starch film in this work was very brittle. It could not be cut to the desired shape for tensile testing. It has been reported that dispersed thermoplastic starch (TPS) phases act as defect sites to decrease the tensile properties of the PLLA film matrices [32].

All the tensile curves of the PLLA-PEG-PLLA/starch films in Figure 8 (below) exhibited a yield point that suggested that they were flexible. The tensile stresses and Young’s moduli of the films slightly decreased, and the strain at break largely decreased, as the starch content increased. It is well known that the tensile properties of starch materials are poor. From Table 3, it is seen that the tensile properties of all the PLLA/starch and PLLA-PEG-PLLA/starch films were lower than those of the pure PLLA and PLLA-PEG-PLLA films, respectively. Therefore, the decreasing tensile properties of both the PLLA/starch and PLLA-PEG-PLLA/starch film types were due to the poor tensile properties of starch and followed the “rule of mixtures” [32,33]. However, all the PLLA-PEG-PLLA/starch films still had higher strain at break (14.4–60.6%) than that of the pure PLLA film (3.4%). Thus, the more flexible PLLA-PEG-PLLA/starch composites have broader applications, such as flexible packaging, etc., than the pure PLLA and PLLA/starch composites.

### 3.5. Water Contact Angle and Moisture Uptake

The hydrophilicity of film samples was determined from their water contact angles, as shown in Figure 9 and also summarized in Table 3. The water contact angles of pure PLLA and PLLA-PEG-PLLA films were 81.5° and 68.9°, respectively, implying that the hydrophilicity of PLLA-PEG-PLLA was higher than that of PLLA because of the hydrophilic PEG middle blocks. The water contact angles of the composite films decreased steadily (hydrophilicity increased) as the starch content increased. This was due to the high hydrophilicity of starch phases [15,20].

The moisture uptake of the film samples was also determined for 48 h, as presented in Figure 10. The moisture uptake of pure PLLA and PLLA-PEG-PLLA films at 48 h was 7.7% and 0.6%, respectively, meaning that PLLA-PEG-PLLA was more hydrophilic than PLLA [14,34] corresponding to the results for the water contact angle. The moisture uptake of both the PLLA and PLLA-PEG-PLLA composite films increased as the starch content increased. The results of water contact angle and moisture uptake suggest that the PLLA-PEG-PLLA/starch composites had higher hydrophilicity than the PLLA/starch composites. In our previous work [14], PLLA/TPS and PLLA-PEG-PLLA/TPS composites were found to have higher hydrophilicity than the PLLA/native starch and PLLA-PEG-PLLA/native starch composites for the same blend ratio. This may be explained by the PLLA/TPS blends having smaller TPS particles in their film matrices than did native starch particles, whereas PLLA-PEG-PLLA/TPS blends were of the co-continuous phase type.

## 4. Conclusions

In this study, the effect of native cassava starch on the properties of PLLA-based and PLLA-PEG-PLLA-based composites was investigated. For the PLLA-PEG-PLLA/starch composites, improved thermal stability of PLLA-PEG-PLLA matrices and good compatibility between PLLA-PEG-PLLA and starch were observed compared to PLLA/starch composites. The PLLA-T_d,max_ peaks of PLLA-PEG-PLLA at 319 °C dramatically shifted to higher temperatures by 17 °C, 30 °C and 44 °C when the starch content was 5, 10 and 20 wt%, respectively, but PLLA did not shift, as determined from TGA. The tensile properties of both the PLLA/starch and PLLA-PEG-PLLA/starch films decreased as the native starch content increased. However, all the PLLA-PEG-PLLA/starch films still exhibited a yield point and showed higher extensibility than the pure PLLA film. The hydrophilicity of both PLLA and PLLA-PEG-PLLA increased with the addition of native starch, as revealed by investigation of their water contact angles and moisture uptake.

The flexible PLLA-PEG-PLLA/starch composites, with balanced thermal stability, mechanical properties, hydrophilicity and cost-effectiveness, are very promising for the development of fully biodegradable biomedical, packaging and agricultural applications.

## Figures and Tables

**Figure 1 polymers-14-03186-f001:**
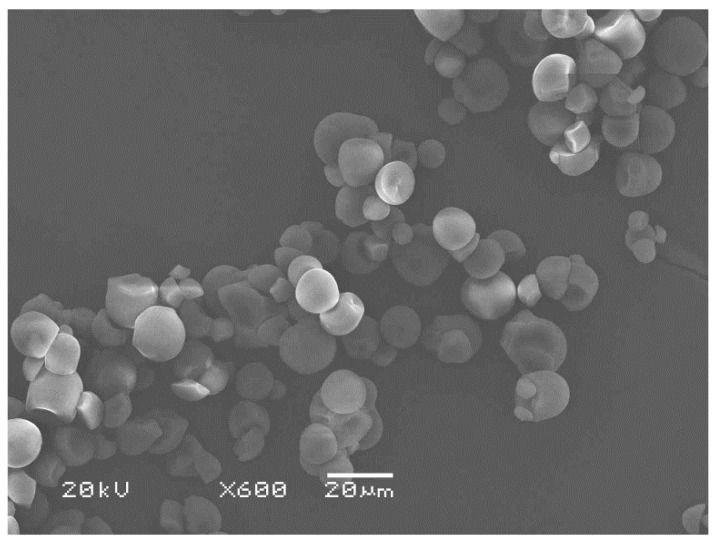
SEM image of cassava starch (bar scale = 20 µm).

**Figure 2 polymers-14-03186-f002:**
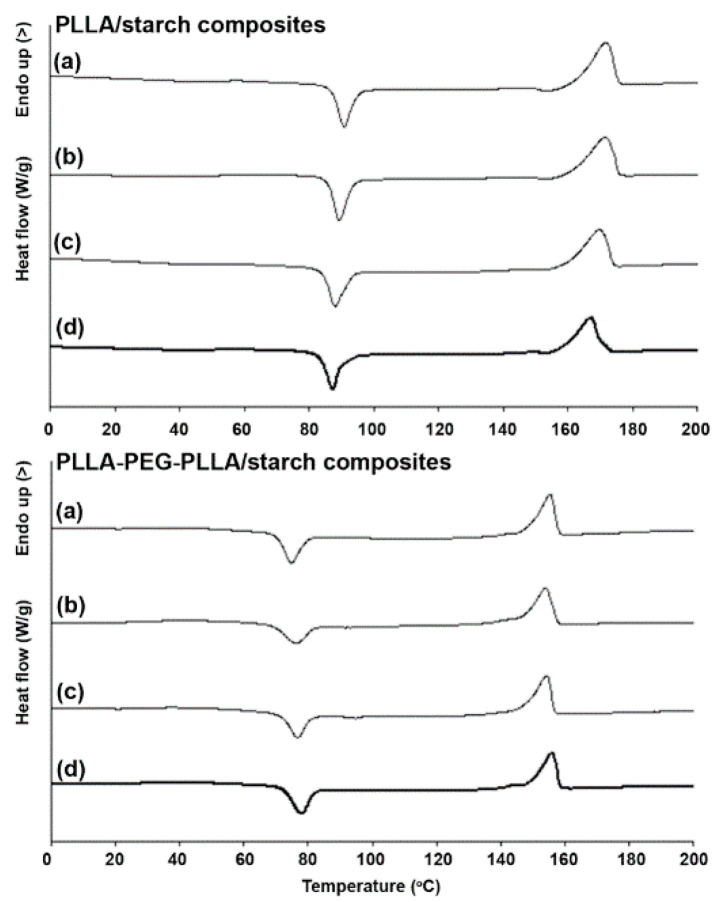
DSC heating curves of (**above**) PLLA/starch and (**below**) PLLA-PEG-PLLA/starch composites with blend ratios of (**a**) 100/0, (**b**) 95/5, (**c**) 90/10 and (**d**) 80/20 (*w*/*w*).

**Figure 3 polymers-14-03186-f003:**
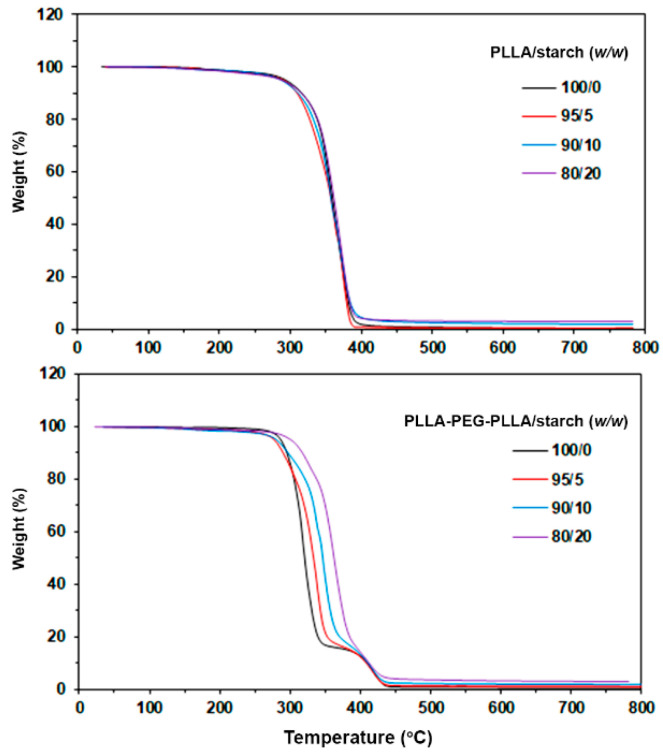
TG thermograms of (**above**) PLLA/starch and (**below**) PLLA-PEG-PLLA/starch composites with various blend ratios.

**Figure 4 polymers-14-03186-f004:**
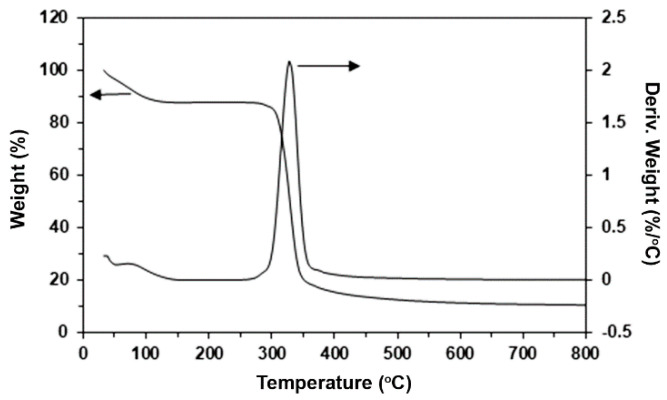
TG and DTG thermograms of native cassava starch.

**Figure 5 polymers-14-03186-f005:**
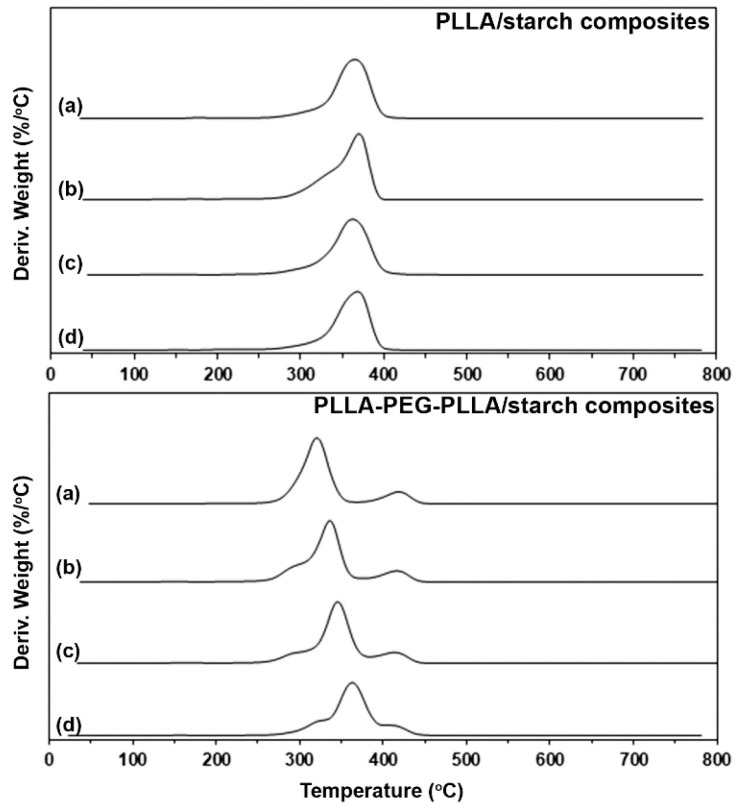
DTG thermograms of (**above**) PLLA/starch and (**below**) PLLA-PEG-PLLA/starch composites with blend ratios of (**a**) 100/0, (**b**) 95/5, (**c**) 90/10 and (**d**) 80/20 (*w*/*w*).

**Figure 6 polymers-14-03186-f006:**
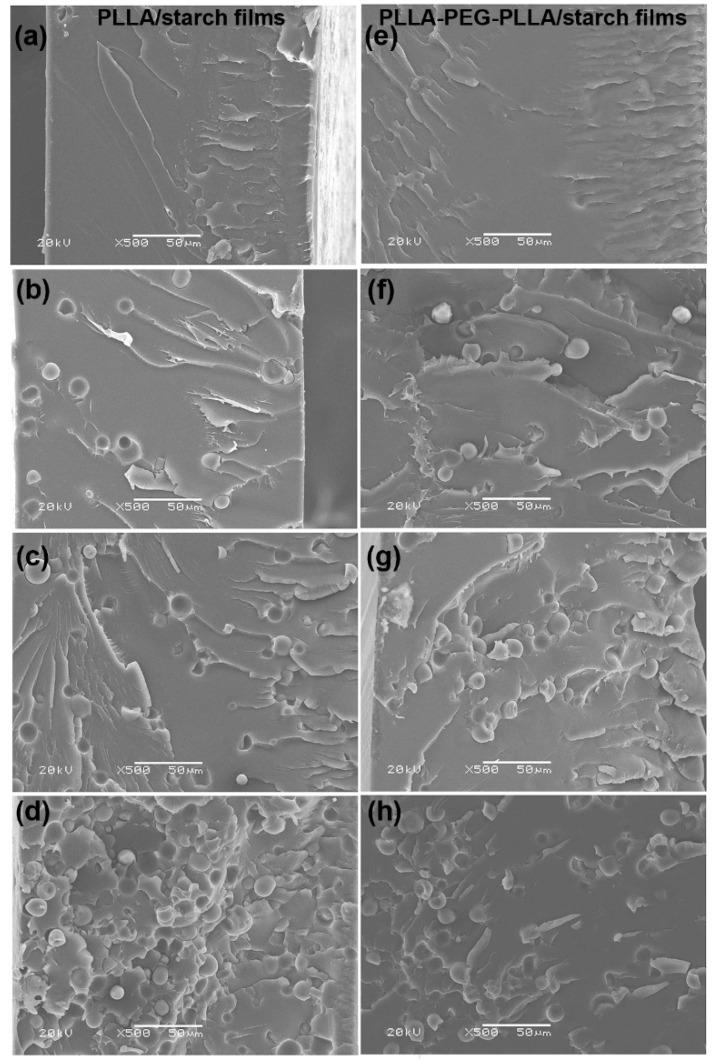
SEM images of cryogenically fractured surfaces of (**a**) pure PLLA film and PLLA/starch films with blend ratios of (**b**) 95/5, (**c**) 90/10 and (**d**) 80/20 (*w*/*w*), as well as (**e**) pure PLLA-PEG-PLLA film and PLLA-PEG-PLLA/starch films with blend ratios of (**f**) 95/5, (**g**) 90/10 and (**h**) 80/20 (*w*/*w*) (all bar scales = 50 µm).

**Figure 7 polymers-14-03186-f007:**
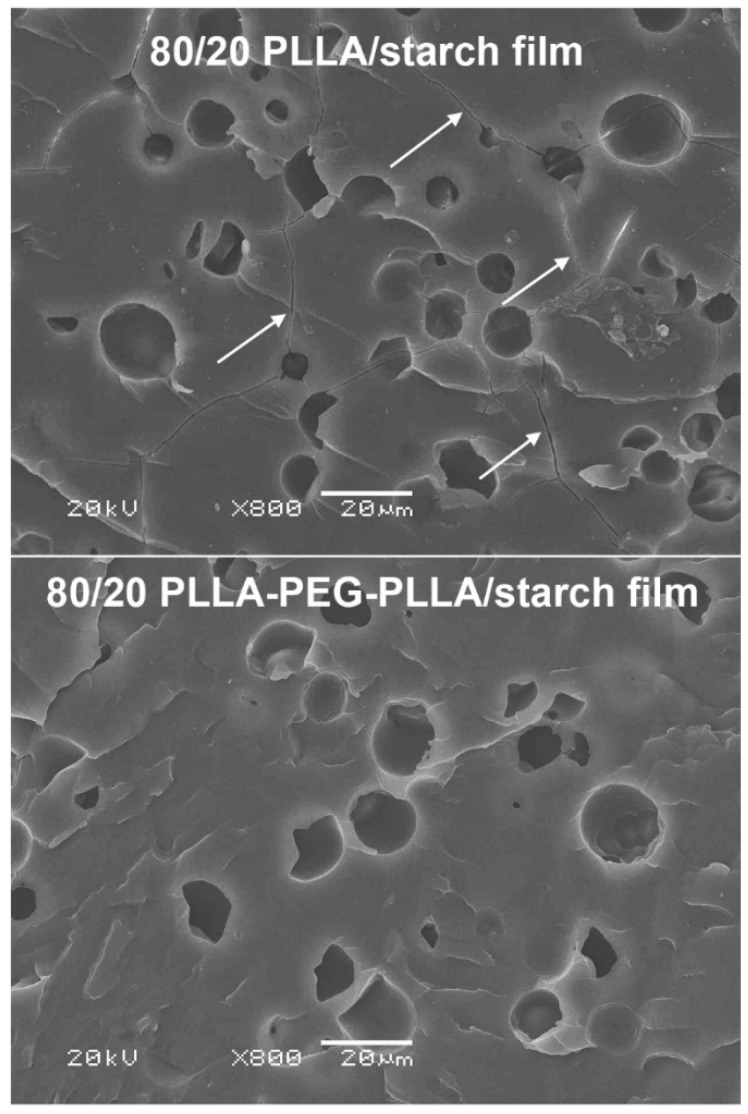
SEM images of cryogenically fractured surfaces for (**above**) 80/20 PLLA/starch and (**below**) 80/20 PLLA-PEG-PLLA/starch films after immersion in 6 N HCl solution for 3 h (some crack surfaces are indicated by white arrows; all bar scales = 20 μm).

**Figure 8 polymers-14-03186-f008:**
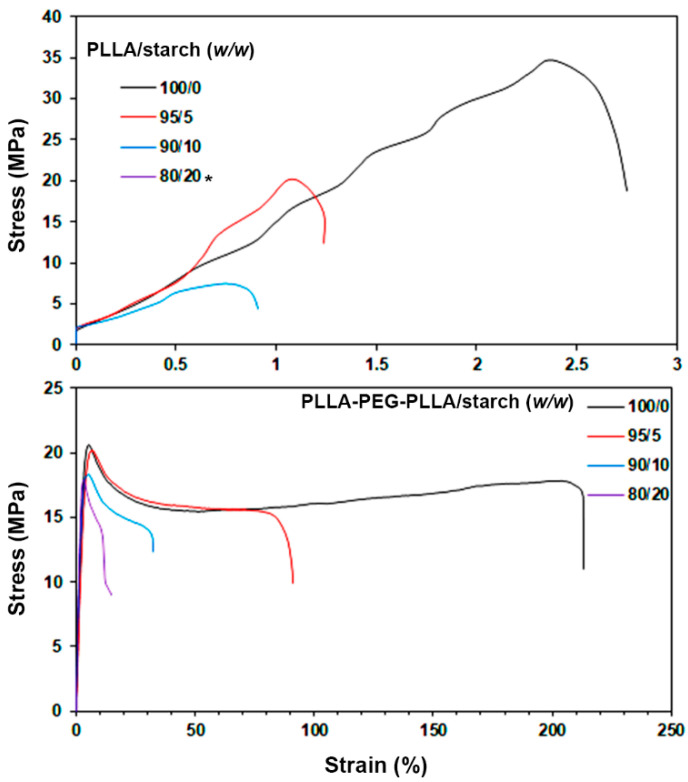
Selected tensile curves of (**above**) PLLA/starch and (**below**) PLLA-PEG-PLLA/starch films with various blend ratios (* 80/20 PLLA/starch film was not determined).

**Figure 9 polymers-14-03186-f009:**
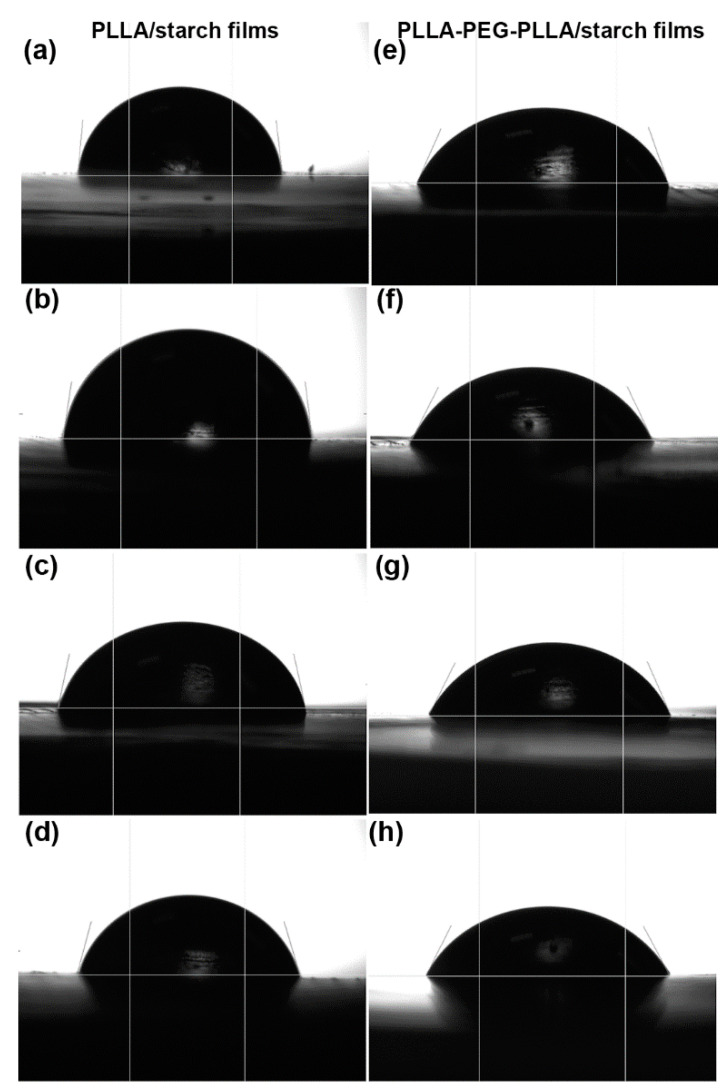
Images of water contact angles of (**a**) pure PLLA film and PLLA/starch films with blend ratios of (**b**) 95/5, (**c**) 90/10 and (**d**) 80/20 (*w*/*w*), as well as (**e**) pure PLLA-PEG-PLLA film and PLLA-PEG-PLLA/starch films with blend ratios of (**f**) 95/5, (**g**) 90/10 and (**h**) 80/20 (*w*/*w*).

**Figure 10 polymers-14-03186-f010:**
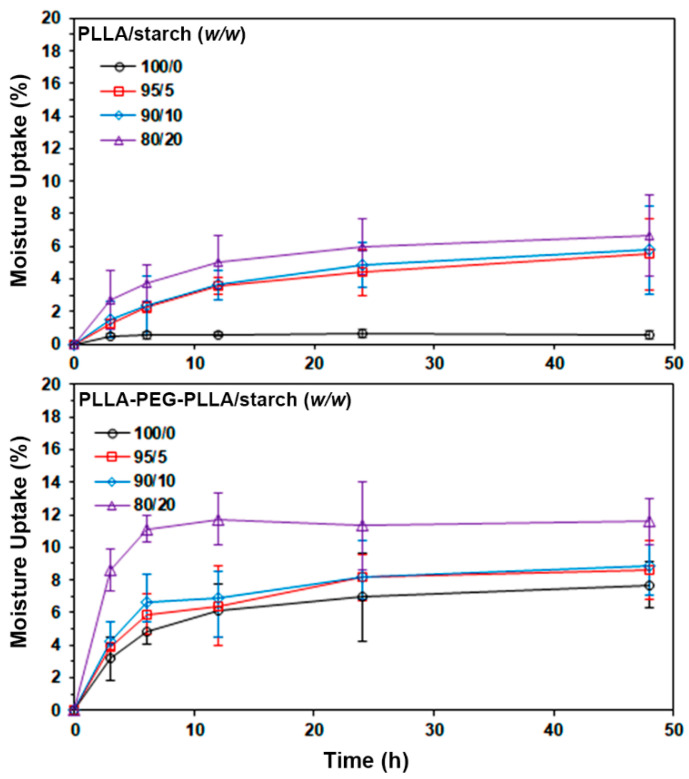
Moisture uptake of (**above**) PLLA/starch and (**below**) PLLA-PEG-PLLA/starch films with various blend ratios.

**Table 1 polymers-14-03186-t001:** Thermal transition properties of PLLA/starch and PLLA-PEG-PLLA/starch composites obtained from DSC heating curves in Figure 2.

Sample	T_g_ (°C) ^a^	T_cc_ (°C) ^b^	ΔH_cc_ (J/g) ^c^	T_m_ (°C) ^d^	ΔH_m_ (J/g) ^e^	*X*_c_ (%) ^f^
PLLA/starch (*w*/*w*)						
100/0	53	90	29.5	172	51.6	23.6
95/5	53	90	29.6	172	53.7	27.1
90/10	52	88	29.5	170	48.5	22.6
80/20	52	88	26.2	168	40.2	18.7
PLLA-PEG-PLLA/starch (*w*/*w*)						
100/0	32	76	17.9	155	29.8	15.3
95/5	32	78	14.7	154	29.7	20.3
90/10	32	77	15.0	154	28.6	19.4
80/20	32	78	16.6	155	26.6	16.1

^a^ Glass transition temperature (T_g_); ^b^ Cold crystallization temperature (T_cc_); ^c^ Enthalpy of cold crystallization (ΔH_cc_); ^d^ Melting temperature (T_m_); ^e^ Enthalpy of melting (ΔH_m_); ^f^ Degree of crystallinity (*X*_c_) calculated from Equation (1).

**Table 2 polymers-14-03186-t002:** Residue ash and temperature of maximum decomposition rate (T_d,max_) of the composites.

Sample	5%-T_d_ (°C) ^a^	Residue ash (wt%) ^b^	PLLA-T_d,max_ (°C) ^c^	PEG-T_d,max_ (°C) ^d^
PLLA/starch (*w*/*w*)				
100/0	291	0.06	365	-
95/5	291	0.59	369	-
90/10	287	1.64	366	-
80/20	285	2.61	369	-
PLLA-PEG-PLLA/starch (*w*/*w*)				
100/0	286	0.11	319	421
95/5	278	0.79	336	421
90/10	280	1.74	349	418
80/20	300	3.05	363	419

^a^ Decomposition temperature at 5% weight loss obtained from TG thermograms in Figure 3; ^b^ Obtained from TG thermograms at 800 °C in Figure 3; ^c^ Temperature of maximum decomposition rate for PLLA blocks (PLLA-T_d,max_) obtained from DTG thermograms in Figure 5; ^d^ Temperature of maximum decomposition rate for PEG blocks (PEG-T_d,max_) obtained from DTG thermograms in Figure 5.

**Table 3 polymers-14-03186-t003:** Tensile properties and water contact angles of PLLA/starch and PLLA-PEG-PLLA/starch films.

Film Samples	Ultimate Tensile Stress (MPa)	Strain at Break (%)	Young’s Modulus (MPa)	Water Contact Angle (°)
PLLA/starch (*w*/*w*)				
100/0	38.4 ± 3.1	3.4 ± 0.7	957 ± 57	81.5 ± 3.3
95/5	24.6 ± 2.5	1.8 ±0.4	782 ±68	81.1 ± 4.1
90/10	6.2 ± 2.2	0.8 ± 0.5	229 ± 62	77.8 ± 3.8
80/20	- *	- *	- *	75.1 ± 2.7
PLLA-PEG-PLLA/starch (*w*/*w*)				
100/0	20.4 ± 1.8	239.6 ± 15.1	321 ± 31	68.9 ± 4.2
95/5	20.6 ± 1.5	90.6 ± 12.5	251 ± 16	67.5 ± 3.1
90/10	19.8 ± 1.9	31.2 ± 8.2	238 ± 25	64.4 ± 2.5
80/20	17.4 ± 0.8	14.4 ± 5.3	212 ± 12	63.9 ± 3.2

* was not determined because it was very brittle.

## Data Availability

The data presented in this study are available on request from the corresponding author.

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
