# Peer review of "Improvement in Thermal Stability of Flexible Poly(L-lactide)-b-poly(ethylene glycol)-b-poly(L-lactide) Bioplastic by Blending with Native Cassava Starch"

_polymers, 2022, doi:10.3390/polym14153186_

Round 1

Reviewer 1 Report

Page

Comment

1, Paragraph 2

The melt flow properties of PLLA-PEG-PLLA can be controlled by reacting with a chain extender to form long-chain branching structures. Whereas, our previous works have reported that thermal stability of PLLA-PEG-PLLA decreased after the chain-extension reaction – Could you please justify the sentence? There was a contradictory statement found in the sentence.

2, Paragraph 1

Therefore, the aim of this work is to investigate the influence of native cassava starch (5%, 10% and 20% by weight) on thermal stability of PLLA-PEG-PLLA. Phase morphology, mechanical properties and hydrophilicity of the PLLA-PEG-PLLA/native cassava-starch composites were also determined. The PLLA/native cassava-starch composites were also prepared by the same method for comparison. – Please rearrange and simplify the sentence in a better manner.

2, Section 2.1

Therefore, the PLLA-PEG-PLLA was chain-extended using 4 parts per hundred of resin by weight (phr) of Joncryl® ADR4368 (BASF, Bangkok, Thailand) as a chain extender to adjust the MFI as 23 g/10 min – Why the value of MFI should be 23 g/10 min? Please provide the reasons behind this.

3, Section 2.3

The tensile testing of film samples (100 mm ´ 10 mm) was carried out using an universal mechanical testing machine (LY-1066B, Dongguan Liyi Environmental Technology Co., Ltd., Guangdong, China) with a 100 kg load cell. A crosshead speed of 50 mm/min and a gauge length of 50 mm were used. The tensile properties were averaged from at least five measurements. – Please state the types of universal standard used for tensile testing.

4, Paragraph 2

The results suggested that the phase compatibility between PLLA-PEG-PLLA and starch was better for nucleation effect. – Please add another sentence for this, it seems incomplete and hanging. Please add more recent and related citations to support the findings.

5, Section 3.2

The TG thermogram of native cassava starch had two steps of thermal decomposition in the ranges 50−150ËšC and 250−500ËšC as presented in Figure S1, due to evaporation of residue moisture and pyrolysis of starch, respectively. – Where is Figure S1 which represents native cassava starch? Please justify.

5, Section 3.2

The residue ashes of both the PLLA/starch and PLLA-PEG-PLLA/starch composites increased steadily with the starch contents as summarized in Table 2. – Please be careful with the font format for this sentence.

6,  Section 3.2

The native cassava starch had a Td,max peak at 327ËšC as assigned in Figure S1. Where is Figure S1 which represents native cassava starch? Please justify.

6,  Section 3.2

However, the PEG-Td,max peaks of the composites were in the range 418−422ËšC that did not shift significantly by starch blending. Please justify the reasons behind this.

7, Section 3.3

 Both the pure PLLA and PLLA-PEG-PLLA films in Figures. 5 (a, left) and 5 (a, right), respectively, had no phase separation. – It would be misleading to put the same Figure 5 a for both images of PLLA and PLLA-PEG-PLLA films. Please use other alphabets.

9, Figure 6

SEM images of cryogenically fractured surfaces for (above) 80/20 PLLA/starch and (below) 80/20 PLLA-PEG-PLLA/starch composite films after immersion in 6 N HCl solution for 3 h (All bar scales = 20 mm). Please pointed out the highlighted images with arrow and explanation to differentiate two types of images. Please label the image separately as well.

9, Section 3.4

The decreasing tensile properties of both the PLLA/starch and PLLAPEG-PLLA/starch films was due to the poor mechanical properties of starch and followed the “rule of mixtures” – Please elaborate with proper citations regarding rule of mixture.

9, Section 3.4

Thus, the more flexible PLLA-PEG-PLLA/starch composites had broader applications than the pure PLLA and PLLA/starch composites. What is the meaning of broader applications for flexible PLLA-PEG-PLLA/starch composites??

10, Section 3.5

The moisture uptake of pure PLLA and PLLA-PEG-PLLA films at 48 h were 7.7% and 0.6%, respectively, meaning that the PLLA-PEG-PLLA was more hydrophilic than the PLLA corresponding to the results of water contact angle. Please provide supporting references for this statement.

11,  Section 3.5

PLLA/TPS and PLLA-PEG-PLLA/TPS composites were found to have more hydrophilicity than the PLLA/native starch and PLLA-PEG-PLLA/native starch composites for the same blend ratio. Where are the images for PLLA/TPS and PLLA-PEG-PLLA/TPS?

4. Conclusions

The hydrophilicity of both the PLLA and PLLA-PEG-PLLA increased with the addition of native starch as reveled by investigation of their water contact angles and moisture uptakes. Please change reveled to revealed.

The flexible PLLAPEG-PLLA/starch composites with balanced thermal stability, mechanical properties, hydrophilicity and cost-effectiveness are very promising for biomedical, packaging and agricultural applications. Please add one subsection for the potential industrial applications of PLLAPEG-PLLA/starch composites with updated references in the manuscript.

Other comments:

It is suggested for the author to include the mechanism behind the incorporation of PLLAPEG-PLLA/starch composites

Author Response

Manuscript ID: polymers-1793832

Title: Improvement Thermal Stability of Flexible Poly(L-lactide)-b-polyethylene glycol-b-poly(L-lactide) Bioplastic by Blending with Native Cassava Starch

Authors: Yaowalak Srisuwan and Yodthong Baimark

The authors would like to thank the reviewers most sincerely for the time that they have spent reading the paper and for their perceptive comments. All of them have been used to improve the paper. A detailed point-by-point set of responses to the reviewer inputs is provided. All corrections are marked in red.

Reviewer # 1

Reviewer # 1: 1/17) Page 1, Paragraph 2:

“The melt flow properties of PLLA-PEG-PLLA can be controlled by reacting with a chain extender to form long-chain branching structures. Whereas, our previous works have reported that thermal stability of PLLA-PEG-PLLA decreased after the chain-extension reaction” – Could you please justify the sentence? There was a contradictory statement found in the sentence.

Authors: Thank you for your comment. The “Whereas” has changed to “However” on P. 1, Paragraph 2 of the revised manuscript.

Reviewer # 1: 2/17) Page 2, Paragraph 1:

“Therefore, the aim of this work is to investigate the influence of native cassava starch (5%, 10% and 20% by weight) on thermal stability of PLLA-PEG-PLLA. Phase morphology, mechanical properties and hydrophilicity of the PLLA-PEG-PLLA/native cassava-starch composites were also determined. The PLLA/native cassava-starch composites were also prepared by the same method for comparison.” – Please rearrange and simplify the sentence in a better manner.

Authors: This sentence was re-written as “Therefore, the aim of this work is to investigate the influence of native cassava starch (5%, 10% and 20% by weight) on thermal stability, phase morphology, mechanical properties and hydrophilicity of the of PLLA-PEG-PLLA compared to the PLLA/native cassava-starch blends.” on P. 2, Paragraph 1 of the revised manuscript.

Reviewer # 1: 3/17) Page 2, Section 2.1:

“Therefore, the PLLA-PEG-PLLA was chain-extended using 4 parts per hundred of resin by weight (phr) of Joncryl® ADR4368 (BASF, Bangkok, Thailand) as a chain extender to adjust the MFI as 23 g/10 min” – Why the value of MFI should be 23 g/10 min? Please provide the reasons behind this.

Authors: MFI of synthesized PLLA-PEG-PLLA was adjusted to 23 g/10 min by chain-extension reaction to near the value obtained with PLLA (24 g/10 min) as described on P. 2, Section 2.1 of the revised manuscript.

Reviewer # 1: 4/17) Page 3, Section 2.3:

“The tensile testing of film samples (100 mm ´ 10 mm) was carried out using an universal mechanical testing machine (LY-1066B, Dongguan Liyi Environmental Technology Co., Ltd., Guangdong, China) with a 100 kg load cell. A crosshead speed of 50 mm/min and a gauge length of 50 mm were used. The tensile properties were averaged from at least five measurements.” – Please state the types of universal standard used for tensile testing.

Authors: Type of universal standard for tensile testing in this work is ASTM D882 as described on P. 3, Section 2.3 of the revised manuscript.

Reviewer # 1: 5/17) Page 4, Paragraph 2:

“The results suggested that the phase compatibility between PLLA-PEG-PLLA and starch was better for nucleation effect.” – Please add another sentence for this, it seems incomplete and hanging. Please add more recent and related citations to support the findings.

Authors: Other sentences with related citations were added on P. 4, Paragraph 2 of the revised manuscript.

Reviewer # 1: 6/17) Page 5, Section 3.2:

“The TG thermogram of native cassava starch had two steps of thermal decomposition in the ranges 50−150ËšC and 250−500ËšC as presented in Figure S1, due to evaporation of residue moisture and pyrolysis of starch, respectively.” – Where is Figure S1 which represents native cassava starch? Please justify.

Authors: TG and DTG thermograms of native cassava starch are presented as Figure 4 on P. 6 of the revised manuscript.

Reviewer # 1: 7/17) Page 5, Section 3.2:

“The residue ashes of both the PLLA/starch and PLLA-PEG-PLLA/starch composites increased steadily with the starch contents as summarized in Table 2.” – Please be careful with the font format for this sentence.

Authors: The font format of this sentence has carefully checked as shown on P. 5, Section 3.2 of the revised manuscript.

Reviewer # 1: 8/17) Page 6, Section 3.2:

“The native cassava starch had a Td,max peak at 327ËšC as assigned in Figure S1.” Where is Figure S1 which represents native cassava starch? Please justify.

Authors: TG and DTG thermograms of native cassava starch are presented as Figure 4 on P. 6 of the revised manuscript.

Reviewer # 1: 9/17) Page 6, Section 3.2:

“However, the PEG-Td,max peaks of the composites were in the range 418−422ËšC that did not shift significantly by starch blending.” Please justify the reasons behind this.

Authors: The reason for this result was “This may be due to starch fractions being completely decomposed before thermal decomposition of PEG” as described on P. 6, Paragraph 2 of the revised manuscript.

Reviewer # 1: 10/17) Page 7, Section 3.3:

“Both the pure PLLA and PLLA-PEG-PLLA films in Figures. 5 (a, left) and 5 (a, right), respectively, had no phase separation.” – It would be misleading to put the same Figure 5 a for both images of PLLA and PLLA-PEG-PLLA films. Please use other alphabets.

Authors: Figure 6 and its caption have re-written as shown on Page 9 of the revised manuscript. Alphabets of (a)-(d) and (e)-(h) were used for PLLA/starch and PLLA-PEG-PLLA film series, respectively.

Reviewer # 1: 11/17) Page 9, Figure 6:

“SEM images of cryogenically fractured surfaces for (above) 80/20 PLLA/starch and (below) 80/20 PLLA-PEG-PLLA/starch composite films after immersion in 6 N HCl solution for 3 h (All bar scales = 20 mm).” Please pointed out the highlighted images with arrow and explanation to differentiate two types of images. Please label the image separately as well.

Authors: The caption of Figure 6 has been re-written as “Figure 7. SEM images of cryogenically fractured surfaces for (above) 80/20 PLLA/starch and (below) 80/20 PLLA-PEG-PLLA/starch films after immersion in 6 N HCl solution for 3 h (some crack surfaces were indicated by white arrows, All bar scales = 20 µm).” on P. 10, of the revised manuscript. This figure has referred with re-written sentence on P. 9.

Reviewer # 1: 12/17) Page 9, Section 3.4:

“The decreasing tensile properties of both the PLLA/starch and PLLAPEG-PLLA/starch films was due to the poor mechanical properties of starch and followed the “rule of mixtures”” – Please elaborate with proper citations regarding rule of mixture.

Authors:  The “rule of mixture” has been elaborated upon in P. 10, Paragraph 2 of the revised manuscript.

Reviewer # 1: 13/17) Page 9, Section 3.4:

“Thus, the more flexible PLLA-PEG-PLLA/starch composites had broader applications than the pure PLLA and PLLA/starch composites.” What is the meaning of broader applications for flexible PLLA-PEG-PLLA/starch composites??

Authors: The applications is such as flexible packaging etc. on P. 11 of the revised manuscript.

Reviewer # 1: 14/17) Page 10, Section 3.5:

“The moisture uptake of pure PLLA and PLLA-PEG-PLLA films at 48 h were 7.7% and 0.6%, respectively, meaning that the PLLA-PEG-PLLA was more hydrophilic than the PLLA corresponding to the results of water contact angle.” Please provide supporting references for this statement.

Authors: Supporting references (ref. nos. 14 and 34) were added on P. 12 of the revised manuscript.

Reviewer # 1: 15/17) Page 11,  Section 3.5:

“PLLA/TPS and PLLA-PEG-PLLA/TPS composites were found to have more hydrophilicity than the PLLA/native starch and PLLA-PEG-PLLA/native starch composites for the same blend ratio.” Where are the images for PLLA/TPS and PLLA-PEG-PLLA/TPS?

Authors: Images of water contact angles of the film samples are illustrated as Figure 9 on P.12 of the revised manuscript.

Reviewer # 1: 16/17) Page 4. Conclusions:

“The hydrophilicity of both the PLLA and PLLA-PEG-PLLA increased with the addition of native starch as reveled by investigation of their water contact angles and moisture uptakes.” Please change reveled to revealed.

The flexible PLLAPEG-PLLA/starch composites with balanced thermal stability, mechanical properties, hydrophilicity and cost-effectiveness are very promising for biomedical, packaging and agricultural applications. Please add one subsection for the potential industrial applications of PLLAPEG-PLLA/starch composites with updated references in the manuscript.

Authors: Subsection for applications of PLLA-PEG-PLLA/starch composites has been added on P.13 in the Conclusion part of the revised manuscript.

Reviewer # 1: 17/17)

It is suggested for the author to include the mechanism behind the incorporation of PLLAPEG-PLLA/starch composites

Authors: Influence of starch blending on properties of PLLA-PEG-PLLA has been referred to in Sections 3.2 and 3.3. PEG middle-blocks significantly changed thermal decomposition behaviors and phase compatibility of the composites compared to PLLA/starch composites.

Reviewer 2 Report

1. In DSC results, the starch exhbited an endo peak due to the mosiure evaporation. Such peak might interpret the cold crystalization peak and thereby influence the calcuation of crystallinity. I strongly recommend that the authors provide the DSC curves of pure starch and compare them. 

2. In TGA tests, please also provide the TG and DTG curves of pure starch for comparison. In addition, please provide the Tonset (intial decomposition temperture) and compare. 

3. In Table 3, the mechanical data of PLLA/starch (tensile strength) is different from the stress-strain curve in Figure 7. Please double check them seriously.

4. In Table 1, please unify the font size.

Thanks. 

Author Response

Reviewer # 2

Reviewer # 2: 1/4) In DSC results, the starch exhbited an endo peak due to the mosiure evaporation. Such peak might interpret the cold crystalization peak and thereby influence the calcuation of crystallinity. I strongly recommend that the authors provide the DSC curves of pure starch and compare them. 

Authors: We have determined DSC heating curve of native starch as below. This has no endo-peak of moisture evaporation. This may be due to the sample being heated at 200ËšC for 3 min to erase the thermal history before fast quenching to 0ËšC and heating from 0ËšC to 200ËšC at a rate of 10ËšC/min under a nitrogen gas flow as described on P. 3, Section 2.3. The moisture could evaporate during heating at 200ËšC for 3 min.

Figure 1. DSC heating curves of native cassava starch.

Reviewer # 2: 2/4)
In TGA tests, please also provide the TG and DTG curves of pure starch for comparison. In addition, please provide the Tonset (initial decomposition temperature) and compare. 

Authors: The 5%-Td values of PLLA/starch and PLLA-PEG-PLLA/starch from TG thermograms have been added in Table 2 on P.6 and are described on P. 5, Section 3.2 of the revised manuscript.

Reviewer # 2: 3/4)

In Table 3, the mechanical data of PLLA/starch (tensile strength) is different from the stress-strain curve in Figure 7. Please double check them seriously.

Authors: Thank you for your comment. We have re-checked and corrected the average ultimate tensile stress as reported in Table 3 on P. 11 of the revised manuscript.

Reviewer # 2: 4/4) In Table 1, please unify the font size.

Authors: Font size in Table 1 has been re-written as the same font size.

Round 2

Reviewer 2 Report

The authors have well addressed my concerns. 

Author Response

Point 1: The authors have well addressed my concerns. 

Response 1: Thanks very much for taking your time to review this manuscript. We really appreciate your positive feedback and suggestion.
